# OpenReview forum: "Visual Cues-Induced Jailbreak Attack on Large Vision-Language Models"
_ICLR.cc/2026/Conference — ICLR 2026 Conference Withdrawn Submission_

### Official Review · Reviewer_B8yN · 2025-10-21

**Soundness:** 3
**Presentation:** 3
**Contribution:** 3
**Rating:** 6
**Confidence:** 4

**Summary:**

This paper studies jailbreak attack on large vision-languange models. Existing black-box attack methods tend to hide harmful semantics from text input. Based on the observation that responses generated by LVLMs are susceptible to visual modality information, the authors show that by using visual cues to induce LVLMs, it is possible to effectively bypass the safety guardrails of LVLMs while explicitly including harmful instructions in the text input. Experiments on six open-source models and three commercial closed-source models demonstrate its effectiveness.

**Strengths:**

- This paper studies an important topic of jailbeak attack on LVLMs. It has drawn more and more attention due to popularity of vision-language foundation models.
- Different from previous works focusing on manipulating text input only, this paper proposes to use visual cues and demonstrates its effectiveness without hiding any harmful semantics. This observation is helpful to researchers in this field to develop more effective attacking methods and understanding the vulnerability of LVLMs in visual input.
- The proposed attack method is black-box, thus can be used to different models including close-source models.
- The authors did extensive experiments on nine models. Ablation study on visual cues utilization instruction and analysis on model training method towards safety guardrails are helpful.

**Weaknesses:**

- Since the proposed method does not hide harmful semantics, it is more prone to be detected by safety gaurdrails when image is not considered.
- The method has a step of generating base image, leading to additional cost. The quality of the generated image may also affect the attack effectiveness. When T2I model cannot generate image for some particular question, the attack will fail.

**Questions:**

- In the proposed method, harmful contents are not hidden from text input. If a similar strategy like previous works is used to hide harmful contents, will VCI still be helpful to the attack?

- When creating images with character identities through typography, the demonstration example uses two words. Will it be helpful to include more detailed descriptive words? Could the authors show a set of commonly used identities?

- In 4.4 Table 4, VCI shows better ASR than Q4. Have the authors tried other images and observed how images quality is related to ASR?

---

> ### Author Response · Authors · 2025-11-20
>
> ### Response to question 1, question2, and question 3
> Dear reviewer, we thank you for your positive and constructive feedback. Our answers to each of your questions are as follows:
>
> >**Q1**: In the proposed method, harmful contents are not hidden from text input. If a similar strategy like previous works is used to hide harmful contents, will VCI still be helpful to the attack?
>
> **A**: Similar to [1], we use "(  )" to mask the keywords of harmful questions to hide harmful content (VCI-H). **Experimental results indicate that the strategy of hiding harmful content actually weakens the effectiveness of VCI.** For example, after masking harmful keywords, the ASR of VCI on Qwen-VL dropped from 74.4% to 52.8%. This is because the core idea of VCI is not to deceive LVLMs by disguising harmful questions as benign ones through hiding harmful content, but rather to leverage visual information to directly influence LVLMs' responses. The operation of hiding harmful content may cause LVLMs to fail to correctly parse users' actual intentions due to the models' weak instruction comprehension ability, thereby weakening the attack effectiveness. In Section 4.5 of our paper, we statistically analyze the proportion of failure causes for existing jailbreak attacks. A considerable portion of failures in existing methods stems from the weak instruction comprehension ability of models, which results in their failure to parse users’ actual questions.
>
> | Models       | VCI   | VCI-H          |
> | ------------ | ----- | -------------- |
> | InternVL-2.5 | 79.0% | 74.0% (-5.0%)  |
> | Qwen-VL      | 74.4% | 52.8% (-21.6%) |
> | DeepSeek-VL  | 66.8% | 41.6% (-25.2%) |
>
> [1]Mm-safetybench: A benchmark for safety evaluation of multimodal large language models.
>
> ---
>
> >**Q2**: When creating images with character identities through typography, the demonstration example uses two words. Will it be helpful to include more detailed descriptive words? Could the authors show a set of commonly used identities?
>
> **A**: The identities we use correspond to the harmful questions. Under the "Hate Speech" topic, the "Extremist" identity is more common, while under the "Malware Generation" topic, the "Hacker" identity is more common. **In the supplementary materials we provide, you can see the corresponding character identities we have generated for each question.**
>
> We conduct an experiment in which we additionally embed the more detailed descriptive words you mentioned into the images (VCI-M). Specifically, we embed the topic of each harmful question as a more detailed descriptive word into the images. For example, the cues embedded in the original images are 'Organ Trafficker' and 'Illegal', while now they are 'Organ Trafficker', 'Illegal', and 'Illegal Activity'. For another example, the cues embedded in the original images are 'Bigot' and 'mock', while now they are 'Bigot', 'mock', and 'Hate Speech'. **The experimental results demonstrate that embedding more detailed descriptive words into images has little effect on VCI's performance.** We have supplemented the experimental results of this part in the revised submission (see Section 4.5).
>
> | Models       | VCI   | VCI-M         |
> | ------------ | ----- | ------------- |
> | InternVL-2.5 | 79.0% | 80.4% (+1.4%) |
> | Qwen-VL      | 74.4% | 72.4% (-2.0%) |
> | DeepSeek-VL  | 66.8% | 69.2% (+2.4%) |
> | Average      | 73.4% | 74.0% (+0.6%) |
>
> ---
>
> >**Q3**: In 4.4 Table 3, VCI shows better ASR than Q4. Have the authors tried other images and observed how images quality are related to ASR?
>
> **A**: The higher ASR of VCI compared to that of Q4 may be attributed to the presence of negative character imagery in VCI's image inputs. Prior research [2] shows that harmful images can increase the likelihood of LVLMs generating harmful responses to harmful queries.
>
> To further investigate how images are related to ASR, we conduct the following three experiments:
>     1.Replacing the base negative character image in VCI with a positive character image (VCI-P).
>     2.Replacing the base negative character image in VCI with a neutral ordinary character image (VCI-N).
>     3.Adding Gaussian noise (Standard deviation=50, mean=0) to VCI's images (VCI-Noise).
> The results reveal that harmful images help improve the effectiveness of VCI, while the addition of Gaussian noise has little effect on VCI's performance.
>
> | Models       | VCI   | VCI-P          | VCI-N          | VCI-Noise     |
> | ------------ | ----- | -------------- | -------------- | ------------- |
> | InternVL-2.5 | 79.0% | 75.0% (-4.0%)  | 73.4% (-5.6%)  | 81.0% (+2.0%) |
> | Qwen-VL      | 74.4% | 68.4% (-6.0%)  | 63.6% (-10.8%) | 68.8% (-6.0%) |
> | DeepSeek-VL  | 66.8% | 56.0% (-10.8%) | 63.4% (-3.4%)  | 68.6% (+1.8%) |
> | Average      | 73.4% | 66.5% (-6.9%)  | 66.8% (-6.6%)  | 72.8% (-0.6%) |
>
> [2] Images are Achilles' heel of alignment: Exploiting visual vulnerabilities for jailbreaking multimodal large language models.

---

> ### Author Response · Authors · 2025-11-20
>
> ### Response to weakness 1 and weakness 2
>
> >**W1**: Since the proposed method does not hide harmful semantics, it is more prone to be detected by safety guardrails when image is not considered.
>
> **A**: The primary goal of jailbreak attacks against LVLMs lies in exploring the challenges   that visual modality integration poses to LVLMs' safety guardrails. **Consequently, such research mainly focuses on exploiting the visual modality, distinguishing this research direction from jailbreak attacks on LLMs that do not consider the visual modality.** Without the visual modality, VCI would degenerate into directly inputting harmful instructions to jailbreak LVLMs, which are typically rejected by the models. Our work, however, strategically leverages the influence of visual modality information to induce the model to respond to textual harmful instructions that should be rejected. This precisely exposes a novel safety vulnerability arising from the integration of visual modality into LVLMs' safety guardrails. In fact, for black box attacks against LVLM, almost all methods cannot be successfully implemented without considering image modality.
>
> ---
>
> >**W2**: The method has a step of generating base image, leading to additional cost. When T2I model cannot generate image for some particular question, the attack will fail.
>
>  **A**: **Our method requires only a single base image generated by the T2I model.** All images corresponding to questions in the dataset are derived from this base image, with character identities and keywords embedded using the Pillow library. Compared to existing methods that generate a separate image for each question, this design reduces the computational cost of invoking the T2I model to a nearly negligible level. Meanwhile, it inherently avoids the risk of attack failure caused by the T2I model's inability to generate images for specific questions, which is also an advantage of our method. In the jailbreak examples provided in our paper's Appendix, you can see that in our method, the image inputs for different questions all originate from the same base image.

---

> > ### Author Response · Authors · 2025-11-27
> >
> > Dear Reviewer,
> >
> > We sincerely appreciate your insightful comments on our paper. In our previous response, we addressed all of your comments point-by-point with detailed explanations and have revised the paper accordingly. We believe these responses have satisfactorily resolved the concerns you raised, and we would greatly appreciate it if you could re-evaluate the paper accordingly.
> > ﻿
> >
> > Should you need any further clarification to facilitate your evaluation, we will respond promptly and fully cooperate.
> > ﻿
> >
> > Thank you again for your valuable time and efforts in advancing this research.

---

### Official Review · Reviewer_M6Dw · 2025-10-30

**Soundness:** 2
**Presentation:** 3
**Contribution:** 2
**Rating:** 2
**Confidence:** 4

**Summary:**

This paper proposes VCI (Visual Cues-Induced Attack), a novel black-box jailbreak attack targeting Large Vision-Language Models (LVLMs). Unlike prior methods that hide harmful intent by embedding it into images, VCI keeps the complete harmful question in text and instead uses visual cues embedded in the image (through typography) to induce the model to generate harmful responses. Experiments across 8 open-source and 4 closed-source LVLMs show average ASRs of 77.0% and 78.5%, respectively, outperforming prior SOTA attacks like FigStep and VRP. The study also explores defenses and ablation results, finding that simple defenses (like noise or perplexity filters) are ineffective. Overall, the paper claims to reveal a previously underexplored vulnerability: that LVLMs’ safety alignment can be bypassed not by obscuring harmful intent, but by exploiting the visual modality’s inductive influence on model behavior.

**Strengths:**

- Novel perspective: The attack does not rely on disguising harmful content, unlike prior visual jailbreaks. Instead, it exploits cross-modal susceptibility, revealing a new failure mode of LVLMs.
- Strong empirical results: Consistent and high ASR across both open- and closed-source models.
- Comprehensive evaluation: Includes ablations, topic-level breakdowns, and defense analysis.

**Weaknesses:**

1. Interpretation of Figure 4: The authors show that embeddings for harmful and benign queries remain separable under VCI, yet the model still produces harmful outputs. This is interpreted as “visual cues overriding safety alignment.” However, when a defender knows this, they might be able to reverse engineer to filter out the harmful queries.
2. Attack design assumptions (redundancy and missing controls):
The paper asserts that including the harmful query in text and image form strengthens the attack. However, this redundancy is questionable.
- If removing harmful text increases ASR (as suggested but not explicitly tested), including it would be unnecessary and potentially counterproductive.
- Conversely, if textual inclusion is required for attack success, that contradicts the claim that visual cues alone induce jailbreak behavior.
An ablation testing removal of textual harmful content would clarify the mechanism and strengthen the paper’s conclusions.
3. Lack of mechanistic analysis:
While the empirical vulnerability is evident, the paper does not probe why visual inputs dominate—e.g., whether this arises from cross-modal attention, gradient alignment, or pretraining data biases.

**Questions:**

1. Clarification: Does Figure 4’s separation of embeddings imply that harmfulness is detectable but not mitigated? Could embedding-space separation be leveraged for defense?
2. Design rationale: Why include harmful queries both in the text and the image? Would typography alone (without textual repetition) still succeed? Did you test removing the harmful text from prompts to isolate the effect of visual cues?
3. Mitigation: Can you discuss strategies to mitigate VCI? For example
- Pretraining/fine-tuning: Incorporating safety-aligned multimodal data, etc.
- Inference-time defenses: Visual-textual consistency checks, etc.

---

> ### Author Response · Authors · 2025-11-20
>
> Dear reviewer, thank you for your comments, but it seems there are some misunderstandings on your part related to the core of our work. The core idea of our paper is to use the influence of visual cues to induce LVLMs to respond to harmful questions in the text modality. If the harmful text is removed, the LVLMs have no idea what the question is (no method can successfully jailbreak the model in this scenario). **Therefore, the Weakness 2 and Question 2 that the reviewer raised do not apply to our method.**
>
> For your other questions, our responses are as follows:
>
> >**Q1**: Does Figure 4’s separation of embeddings imply that harmfulness is detectable but not mitigated? Could embedding-space separation be leveraged for defense?
>
> **A**: The separation of embedding spaces means that the model can effectively distinguish between harmful and benign queries. Typically, this means that the model's own safety guardrails can reject harmful queries. **However, our method proves that even if the embedding space is separable, the model can still generate harmful responses under the induction of visual cues, and this is one of our contributions.**
>
> Additionally, this cannot be used for defense, as the inputs of most jailbreak attacks are inseparable **(see Appendix A.7)** , and defense methods need to be generalizable to diverse attack scenarios, including those with inseparable inputs.
>
> ---
>
> >**Q3**: Can you discuss strategies to mitigate VCI?
>
> **A**: We think that employing Guard LLMs (such as Qwen3 or Llama3 Guard) to conduct harmful content detection on the model’s outputs can effectively mitigate the vast majority of jailbreak attacks, including our own method. However, this comes at a high cost.
>
> ---
>
> >**W1**: When a defender knows this, they might be able to reverse engineer to filter out the harmful queries.
>
> **A**: Dear reviewer, while your point itself is not wrong, isn't that exactly the purpose of our work? Our work aims to reveal potential safety vulnerabilities in LVLMs, thereby providing insights for developing more robust safety guardrails. **For any attack method, once disclosed, researchers can develop defensive measures against it based on its principles**, such as the most direct fine-tuning.
>
> >**W3**: Lack of mechanistic analysis: While the empirical vulnerability is evident, the paper does not probe why visual inputs dominate—e.g., whether this arises from cross-modal attention, gradient alignment, or pretraining data biases.
>
> **A**: **We have provided a reasonable explanation in the paper, and you can see more details in Appendix A.3**. We speculate that it may be related to LVLMs' training methods and training data. LVLMs are typically developed by integrating visual components into pre-trained LLMs and are further fine-tuned on tasks such as image captioning and visual question answering. The labels of such tasks are often highly bound to the image content, which may lead to LVLMs' tendency to rely too much on visual information. Thus, even if the image is blank, the model is still influenced by the visual input when generating responses.

---

> > ### Author Response · Authors · 2025-11-27
> >
> > Dear Reviewer,
> >
> > We sincerely appreciate your insightful comments on our paper. In our previous response, we addressed all of your comments point-by-point with detailed explanations and have revised the paper accordingly. We believe these responses have satisfactorily resolved the concerns you raised, and we would greatly appreciate it if you could re-evaluate the paper accordingly.
> > ﻿
> >
> > Should you need any further clarification to facilitate your evaluation, we will respond promptly and fully cooperate.
> > ﻿
> >
> > Thank you again for your valuable time and efforts in advancing this research.

---

### Official Review · Reviewer_TsXZ · 2025-11-01

**Soundness:** 2
**Presentation:** 3
**Contribution:** 2
**Rating:** 4
**Confidence:** 4

**Summary:**

The paper identifies a new failure mode in LVLM safety: model outputs are highly susceptible to visual cues even when the text explicitly contains a harmful request. Building on this, the authors propose VCI, a black-box jailbreak attack that (i) embeds an attacker-chosen identity and keywords as typographic elements in an image, and (ii) explicitly instructs the LVLM to rely on those visual cues to answer a list-style, paraphrased version of the harmful question. Across 8 open-source LVLMs and 4 commercial LVLMs, VCI reports high attack success rates (ASR) and outperforms prior black-box baselines. The paper also studies defenses (self-reminder, perplexity filter, image noise).

**Strengths:**

1: Previous black-box LVLM jailbreaks primarily “move harm into the image” to bypass text-side safety mechanisms. In contrast, VCI preserves the harmful textual content but leverages visual cue–driven compliance to transform a model’s refusal into a direct response, representing a distinct and insightful attack mechanism.

2: The attack uses readily available components (text-to-image, typography, paraphrasing) and a straightforward prompt template. This design not only facilitates replication but also provides a valuable foundation for developing effective defensive strategies.

**Weaknesses:**

1: The paper lacks a clear analysis of the underlying mechanism behind the proposed attack. In particular, it remains unexplained why the introduction of a seemingly innocuous blank image substantially increases the success rate of jailbreak attempts (as shown in Figures 1 and 2). Similarly, the paper does not provide sufficient insight into why the VCI method is able to bypass the safety guardrails of LVLMs. A deeper examination of these mechanisms, such as how visual cues influence the model’s alignment layers or decision boundaries, would greatly strengthen the paper’s technical contribution and interpretability.

**Questions:**

1: See Weakness

---

> ### Author Response · Authors · 2025-11-20
>
> Thank you for your comments. Our answers to each point of your question are as follows:
>
> >**Q1&W1**: The paper lacks a clear analysis of the underlying mechanism behind the proposed attack. In particular, it remains unexplained why the introduction of a seemingly innocuous blank image substantially increases the success rate of jailbreak attempts (as shown in Figures 1 and 2). Similarly, the paper does not provide sufficient insight into why the VCI method is able to bypass the safety guardrails of LVLMs. A deeper examination of these mechanisms, such as how visual cues influence the model’s alignment layers or decision boundaries, would greatly strengthen the paper’s technical contribution and interpretability.
>
> **A**:  The introduction of a seemingly innocuous blank image substantially increasing the success rate of jailbreak attempts is a phenomenon we have observed. This example, together with the one in Figure 2, indicates that **the outputs of LVLMs are susceptible to the visual information, and this influence can interfere with the LVLMs' safety alignment. Therefore, we propose embedding appropriate cues in images and using these visual cues to induce LVLMs to generate harmful responses, which constitutes the core logic behind the effectiveness of VCI. Our extensive ablation experiments have verified this mechanism by quantifying the correlation between the type and quantity of visual cues and the jailbreak success rate.**
>
> **We have provided a detailed explanation for why blank images can cause jailbreaks in the paper (see Appendix A.3 for more details**). Specifically, we speculate that this may be related to the training methods and training data of LVLMs. LVLMs are typically developed by integrating visual components into pre-trained LLMs and are further fine-tuned on tasks such as image captioning and visual question answering. The labels of such tasks are often closely tied to the image content, which may lead to LVLMs' tendency to over-rely on visual information. Even if the image is blank, the model is still influenced by the visual input when generating responses.
>
> We fully agree with you that a deeper theoretical understanding of how visual cues influence the LVLMs' alignment layers or decision boundaries is a highly insightful and critical research direction. However, as is widely recognized, this type of research remains an extremely challenging endeavor across the entire AI field. Given these inherent difficulties and **our work's primary focus on developing jailbreak methods**, it is not realistic to provide a comprehensive theoretical explanation in a single paper. In fact, when looking back at previous black-box jailbreak research [1,2,3,4], we can find that almost no such comprehensive breakthroughs have been achieved in a single paper.
>
> In future research, we will strive towards this direction.
>
> [1] Distraction is all you need for multimodal large language model jailbreaking.
>
> [2] Jailbreak large vision-language models through multi-modal linkage.
>
> [3] Visual-roleplay: Universal jailbreak attack on multimodal large language models via role-playing image character.
>
> [4] Figstep: Jailbreaking large vision-language models via typographic visual prompts.

---

> > ### Author Response · Authors · 2025-11-27
> >
> > Dear Reviewer,
> >
> >
> > We sincerely appreciate your insightful comments on our paper. In our previous response, we addressed all of your comments point-by-point with detailed explanations and have revised the paper accordingly. We believe these responses have satisfactorily resolved the concerns you raised, and we would greatly appreciate it if you could re-evaluate the paper accordingly.
> > ﻿
> >
> > Should you need any further clarification to facilitate your evaluation, we will respond promptly and fully cooperate.
> > ﻿
> >
> > Thank you again for your valuable time and efforts in advancing this research.

---

### Official Review · Reviewer_tUYe · 2025-11-01

**Soundness:** 3
**Presentation:** 3
**Contribution:** 2
**Rating:** 4
**Confidence:** 4

**Summary:**

This paper proposes a black-box jailbreak attack against Large Vision-Language Models (LVLMs) called Visual Cues-Induced Attack (VCI). The method provides an explicit harmful question in the text modality while simultaneously embedding visual cues, such as negative character identities and keywords, into the image modality. VCI aims to exploit a vulnerability where LVLMs, when processing such inputs, are induced to bypass their safety guardrails and generate harmful content.

**Strengths:**

1. The paper's primary strength is its extensive experimental evaluation, covering 12 popular open-source and closed-source LVLMs and comparing against 7 baseline methods.
2. The VCI method achieves high ASR in the experiments, proving that this specific combination of explicit harmful text and visual cues is an effective attack configuration.

**Weaknesses:**

1. The paper's core motivation—that visual information "overrides" internal knowledge —is built on a flawed argument (Figure 2). The model's behavior in the "1840" example is better explained as instruction-following rather than a fundamental override of its safety/factual training.
2. As detailed in the "Contribution" section, the core mechanism of VCI (embedding cues in images) heavily overlaps with existing methods like FigStep and VRP. The paper fails to clearly articulate a fundamental conceptual innovation beyond an incremental change in the attack setup (i.e., making the text prompt explicitly harmful).
3. The paper does not sufficiently disentangle why the attack succeeds. The high ASR could be induced by the specific content of the visual cues (e.g., "Organ Trafficker"), or it could be that the task structure itself (i.e., "infer based on the image") pushes the model into a less safety-aligned state. The ablation Q2 (blank image) hints at the latter but is not deeply explored.

**Questions:**

1. Given that the prompt explicitly directs the model to answer "Based on the image" , how can the authors justify using this example as evidence of visual cues "overriding" internal knowledge, rather than the model simply "following" a task instruction?
2. FigStep also embeds typographic text (e.g., "Methods to... illegal human organ trade") into the image. What is the fundamental mechanistic difference between VCI embedding "Organ Trafficker" and "illegal" and the prompts used by FigStep? Both seem to leverage the model's visual understanding to introduce harmful concepts.
3. How can the authors prove that the attack's success is due to the specific content of the visual cues (e.g., "Organ Trafficker") rather than the task structure itself (i.e., the "infer from image" meta-instruction) disabling the safety guardrails? What is the ASR if benign visual cues (e.g., "Doctor" and "health" in the image) are used with the same harmful text prompt from VCI?

---

> ### Author Response · Authors · 2025-11-20
>
> ### Response to question 1, question 2,  weakness 1, and weakness 2
>
> We appreciate the comments provided by the reviewer and value this opportunity to clarify some misunderstandings. Our answers to the reviewer's questions are as follows:
>
> >**Q1&W1**: Given that the prompt explicitly directs the model to answer "Based on the image" , how can the authors justify using this example as evidence of visual cues "overriding" internal knowledge, rather than the model simply "following" a task instruction? & The paper's core motivation—that visual information "overrides" internal knowledge —is built on a flawed argument (Figure 2). The model's behavior in the "1840" example is better explained as instruction-following rather than a fundamental override of its safety/factual training.
>
> **A**: Dear reviewer, we are sorry that you may have misunderstood the example in Figure 2.
> This example aims to demonstrate the comparative influence of visual versus textual cues on the LVLM's output.
> 1. Scenario 1 (Based on the image): When the LVLM is instructed to answer based on the image containing the text "1840," it outputs the incorrect answer "1840," overriding its internal knowledge that the correct year is 1776.
> 2. Scenario 2 (Based on the text): In contrast, when the LVLM is instructed to answer based only on the same textual statement ("The United States was founded in 1840") provided as raw text, it correctly outputs "1776," disregarding the erroneous textual information.
>
> **This contrast rules out the possibility that the model merely "follows task instructions." If it is just following a task instruction, the LVLM should output "1840" in Scenario 2 (to align with the "based on the text" instruction). Yet it relies on internal knowledge to correct the error.** This example shows that the LVLM is more susceptible to visual cues than to equivalent textual input. Therefore, our method proposes to leverage visual cues to induce LVLMs to generate harmful responses.
>
> ---
>
> >**Q2&W2**: ...the core mechanism of VCI (embedding cues in images) heavily overlaps with existing methods like FigStep and VRP...
>
> **A**: **In Section 5 of our paper, we have fully demonstrated the differences between our method and methods such as FigStep and VRP by analyzing the activation of input prompts in the LVLM** (see original version: lines 429–465 or revised version: lines 441–477). We would like to reiterate here:
> 1. The core logic of prior methods lies in **disguising harmful questions as benign ones to achieve jailbreaks.** In contrast, our method **explicitly inputs complete harmful questions without disguise** while leveraging the influence of visual information to deviate the LVLMs' output from their original refusal responses, ultimately generating harmful content.
> 2. For FigStep and VRP, their typographic visual prompts result in **overlapping embeddings** between benign and harmful queries, which leads to successful jailbreaks. In our method, although the underlying LLM can distinguish between benign and harmful queries (as indicated by the **distinguishable semantic embeddings**), the model ultimately generates harmful content under the influence of visual information.**The visualization of the embeddings for benign and harmful queries under different jailbreak methods is provided in Figure 4 and Figure 10.**
>
> **These fundamental differences in both mechanism and design philosophy distinguish VCI as a substantially novel attack paradigm beyond existing methods.**

---

> ### Author Response · Authors · 2025-11-20
>
> ### Response to question 3 and weakness 3
>
> >**Q3&W3**: How can the authors prove that the attack's success is due to the specific content of the visual cues (e.g., "Organ Trafficker") rather than the task structure itself (i.e., the "infer from image" meta-instruction) disabling the safety guardrails? What is the ASR if benign visual cues (e.g., "Doctor" and "health" in the image) are used with the same harmful text prompt from VCI? & The paper does not sufficiently disentangle why the attack succeeds. The high ASR could be induced by the specific content of the visual cues (e.g., "Organ Trafficker"), or it could be that the task structure itself (i.e., "infer based on the image") pushes the model into a less safety-aligned state. The ablation Q2 (blank image) hints at the latter but is not deeply explored.
>
> **A**: Our ablation experiments with  different embedded visual cues demonstrate that the attack's success is due to the specific content of the visual cues rather than the task structure itself. **The results show that the type and quantity of visual cues have significant impacts on our method's performance. If only the task structure itself (i.e., the "infer from image" meta-instruction) disables the safety guardrails, then when different visual cues are embedded, our method's ASR should remain unchanged**. More details can be found in Section 4.5.
>
> Regarding the question "What is the ASR if benign visual cues (e.g., "Doctor" and "health" in the image) are used with the same harmful text prompt from VCI?", we have supplemented the experiments using benign visual cues (**VCI-B**), and a specific example and more details are available in our revised paper. **The results indicate that our method can still effectively jailbreak LVLMs when benign visual cues are used (Average ASR: 64.2%). However, when harmful visual cues are used, our method's ASR is higher (+9.2%) than that with benign visual cues, which is consistent with our explanation in Section 3.2.** We have supplemented the experimental results of this part in the revised submission (see Section 4.5).
>
> | Models       | VCI   | VCI-B          |
> | ------------ | ----- | -------------- |
> | InternVL-2.5 | 79.0% | 74.0% (-5.0%)  |
> | Qwen-VL      | 74.4% | 65.2% (-9.2%)  |
> | DeepSeek-VL  | 66.8% | 53.4% (-13.4%) |
> | Average      | 73.4% | 64.2% (-9.2%)  |

---

> > ### Author Response · Authors · 2025-11-27
> >
> > Dear Reviewer,
> >
> > We sincerely appreciate your insightful comments on our paper. In our previous response, we addressed all of your comments point-by-point with detailed explanations and have revised the paper accordingly. We believe these responses have satisfactorily resolved the concerns you raised, and we would greatly appreciate it if you could re-evaluate the paper accordingly.
> > ﻿
> >
> > Should you need any further clarification to facilitate your evaluation, we will respond promptly and fully cooperate.
> > ﻿
> >
> > Thank you again for your valuable time and efforts in advancing this research.

---

### Author Response · Authors · 2025-11-26

## Global responses


We sincerely appreciate the time and effort invested by the reviewers and Area Chair. In this rebuttal, we have addressed the reviewers' concerns and clarified potential misunderstandings in detail. We summarize our key responses as follows:

> Clarification on Reviewer tUYe's concerns about "the core mechanism of VCI overlaps with existing methods like FigStep and VRP":

In Section 5 of our paper, we have fully demonstrated the differences between our method and methods such as FigStep and VRP by analyzing the embedding activations of input prompts in the LVLMs (see original version: lines 429–465 or revised version: lines 441–477). We summarize as follows:
1. The core logic of prior methods lies in **disguising harmful questions as benign ones to achieve jailbreaks.** In contrast, our method **explicitly inputs complete harmful questions without disguise** while leveraging the influence of visual information to deviate the LVLMs' output from their original refusal responses, ultimately generating harmful content.
2. For FigStep and VRP, their typographic visual prompts result in **overlapping embeddings** between benign and harmful queries, which leads to successful jailbreaks. In our method, although the underlying LLM can distinguish between benign and harmful queries (as indicated by the **distinguishable semantic embeddings**), the model ultimately generates harmful content under the influence of visual information. **The visualizations of the embeddings for benign and harmful queries under different jailbreak methods are provided in Figure 4 and Figure 10.**

**These fundamental differences in both mechanism and design philosophy distinguish VCI as a substantially novel attack paradigm beyond existing methods.**

---

 >Clarification on Reviewer TsXZ's concerns about "the paper lacks a clear analysis of the underlying mechanism behind the proposed attack. "

In this work, we identify a key safety vulnerability in LVLMs: the outputs of LVLMs are susceptible to the visual information, and **this influence can interfere with the LVLMs' safety alignment. Therefore, we propose embedding appropriate cues in images and using these visual cues to induce LVLMs to generate harmful responses, which constitutes the core logic behind the effectiveness of VCI.**

**Our extensive ablation experiments have verified this mechanism by quantifying the correlation between the type and quantity of visual cues and the jailbreak success rate.** In addition, we have provided a detailed explanation for why blank images can cause jailbreaks in the paper (**see Appendix A.3 for more details**). Considering the limitations of the current research status in the field and our paper's focus on designing jailbreak algorithms, it is challenging to provide a comprehensive theoretical explanation in a single work. We appreciate the reviewer's suggestions and will improve in this direction in the future.

---

> Clarification on Reviewer M6Dw's concerns about "redundancy in attack design":

The core idea of our paper is to use the influence of visual cues to induce LVLMs to respond to harmful questions in the text modality. Specifically:
1. Harmful textual questions are the "attack's target" (i.e., the core content that attackers want LVLMs to answer).
2. Visual cues are the "attack's trigger", which are used to induce LVLMs to generate responses to harmful textual questions.

If the harmful text is removed (as suggested by the reviewer), the LVLMs would have no target question to respond to. This is equivalent to not providing the LVLMs with a question, yet expecting to obtain the LVLMs' answer to this question. No method can successfully jailbreak the model in this scenario. **Therefore, Reviewer M6Dw's concerns do not apply to our method.**

---

>Clarification on Reviewer B8yN's concerns about "the proposed method is more prone to be detected by safety guardrails when the image is not considered":

The primary goal of jailbreak attacks against LVLMs lies in exploring the challenges that visual modality integration poses to LVLMs' safety guardrails. **Consequently, such research mainly focuses on exploiting the visual modality, distinguishing this research direction from jailbreak attacks on LLMs that do not consider the visual modality.** In fact, if the image modality is not considered, almost all jailbreak methods designed for LVLMs will fail.

---

We once again thank the reviewers and Area Chair for their efforts. We believe that all the concerns of the reviewers have been fully addressed, and we kindly request the reviewers to read our response and consider re-evaluating our paper. If the reviewer has any further questions, we are more than happy to discuss them in detail.

---

### Note · Authors · 2026-01-27

I have read and agree with the venue's withdrawal policy on behalf of myself and my co-authors.

---

### Meta-Review · Area_Chair_1yZf · 2026-01-11

**Summary:**

This paper develops VCI, a black-box jailbreak method for LVLMs. Its key strategy is that, by embedding visual cues (keywords/identities) into an image and instructing the model to use them, the model can be induced to answer harmful textual questions that it previously refused. Empirical results with multiple LVLMs are provided to support its effectiveness.

The initial ratings were mixed. While they generally acknowledged the strong empirical results, multiple significant concerns were raised: 1) the novelty over the prior works (e.g., FigStep/VRP) is incremental; 2) it is still unclear (or not sufficiently explained) why this attack works; and 3) some design components in VCI should be better justified.

By reading the rebuttal, the AC believes that this work is sufficiently different from prior works, therefore novelty should not be a major concern. But for the other two points, as detailed in the next part, the AC does not believe they are sufficiently addressed. Therefore, the AC's recommendation is rejection.

**Reviewer Concerns:**

Addressed concern:
1) Novelty: compared to prior works, VCI explicitly keeps the harmful question in the text input. This attacking setup is sufficiently different from prior work.


Outstanding concerns:
1) Why VCI works: This is a common concern shared by multiple reviewers, as the original submission does not sufficiently explain why the visual modality overrides the textual safety alignment. The rebuttal provides plausible speculation, but the explanation remains largely qualitative and does not directly probe cross-modal attention/alignment pathways. Therefore the AC does not believe this concern is sufficiently addressed.

2) Design components in VCI: The authors argue that removing the harmful text removes the “target,” so no jailbreak would happen under this setup. Firstly, the AC agrees that this is a valid argument; but meanwhile, the AC wants to point out that it does not fully address the reviewer’s underlying concern about causal disentanglement and potential redundancy in the attack setup (which again links to the major concern about why VCI works). In other words, the AC believes that an ablation that preserves a clear task while removing explicit harmful text to isolate the independent contribution of visual induction is needed for addressing this concern, which, unfortunately, is not provided in the rebuttal.

**Reviewer Scores:**

Given the outstanding concerns above, the AC believes that the three reviewers, who originally rated negatively on this paper, will not further increase their ratings.

For the reviewer who rated it positively, very likely the score will be maintained or slightly decreased due to these outstanding concerns.

---

### Decision · Program_Chairs · 2026-01-26

Reject